# Relation between Plasma Trough Concentration of Pazopanib and Progression-Free Survival in Metastatic Soft Tissue Sarcoma Patients

**DOI:** 10.3390/pharmaceutics14061224

**Published:** 2022-06-09

**Authors:** Marie-Sophie Minot-This, Pascaline Boudou-Rouquette, Anne Jouinot, Sixtine de Percin, David Balakirouchenane, Nihel Khoudour, Camille Tlemsani, Jonathan Chauvin, Audrey Thomas-Schoemann, François Goldwasser, Benoit Blanchet, Jérôme Alexandre

**Affiliations:** 1Institut du Cancer Paris CARPEM, AP-HP, APHP.Centre, Department of Medical Oncology, ARIANE, Cochin Hospital, 75014 Paris, France; marie-sophie.minot@aphp.fr (M.-S.M.-T.); anne.jouinot@aphp.fr (A.J.); sixtine.depercin@aphp.fr (S.d.P.); camille.tlemsani@aphp.fr (C.T.); francois.goldwasser@aphp.fr (F.G.); jerome.alexandre@aphp.fr (J.A.); 2INSERM U-1016, CNRS UMR-8104, University of Paris, Institut Cochin, 75014 Paris, France; 3Department of Pharmacokinetics and Pharmacochemistry, AP-HP, CARPEM, Cochin Hospital, 75014 Paris, France; david.balakirouchenane@aphp.fr (D.B.); nihel.khoudour@aphp.fr (N.K.); audrey.thomas@aphp.fr (A.T.-S.); benoit.blanchet@aphp.fr (B.B.); 4UMR8038 CNRS, U1268 INSERM, Faculty of Pharmacy, University of Paris, PRES Sorbonne Paris Cité, CARPEM, 75006 Paris, France; 5Lixoft, 92160 Antony, France; jonathan.chauvin@aphp.fr; 6Centre de Recherche des Cordeliers, Université Paris-Sorbonne, INSERM, 75005 Paris, France

**Keywords:** Pazopanib, pharmacokinetics, soft-tissue sarcoma, progression-free survival, toxicity

## Abstract

Background: Pazopanib (PAZ) is an oral angiogenesis inhibitor approved to treat soft tissue sarcoma (STS) but associated with a large interpatient pharmacokinetic (PK) variability and narrow therapeutic index. We aimed to define the specific threshold of PAZ trough concentration (C_min_) associated with better progression-free survival (PFS) in STS patients. Methods: In this observational study, PAZ Cmin was monitored over the treatment course. For the primary endpoint, the 3-month PFS in STS was analyzed with logistic regression. Second, we performed exposure–overall survival (OS) (Cox model plus Kaplan–Meier analysis/log-rank test) and exposure–toxicity analyses. Results: Ninety-five STS patients were eligible for pharmacokinetic/pharmacodynamic (PK/PD) assessment. In the multivariable analysis, PAZ C_min_ < 27 mg/L was independently associated with a risk of progression at 3 months (odds ratio (OR) 4.21, 95% confidence interval (CI) (1.47–12.12), *p =* 0.008). A higher average of PAZ C_min_ over the first 3 months was associated with a higher risk of grade 3–4 toxicities according to the NCI-CTCAE version 5.0 (OR 1.07 per 1 mg/L increase, CI95 (1.02–1.13), *p =* 0.007). Conclusion: PAZ C_min_ ≥ 27 mg/L was independently associated with improved 3-month PFS in STS patients. Pharmacokinetically-guided dosing could be helpful to optimize the clinical management of STS patients in daily clinical practice.

## 1. Introduction

Soft-tissue sarcomas (STS) is a group of rare mesenchymal cancers that includes about 70 histological types and accounts for 1% of adult cancers. In Europe, the estimated yearly incidence is five per 100,000 [1]. The prognosis of metastatic and unresectable stages remains poor and only a slight improvement has been made with the use of doxorubicin and ifosfamide in first-line treatment [2].

Pazopanib (PAZ) is an angiogenesis inhibitor that targets the tyrosine kinase domain of vascular endothelial growth factor receptors 1, 2, and 3; platelet-derived growth factor receptors; and c-kit [3,4]. PAZ is approved for the treatment of advanced renal cell carcinoma (RCC) and chemotherapy-pretreated STS [5]. In the PALETTE trial, PAZ demonstrated a clinical benefit with a longer progression-free survival (PFS) compared to placebo (median PFS 4.6 months (95% CI 3.7–4.8) vs. 1.6 months (0.9–1.8), respectively; hazard ratio (HR) 0.31, *p* < 0.0001), without overall survival improvement [6].

PAZ is administered orally at a flat-fixed dose, despite a large interpatient pharmacokinetic (PK) variability and a low therapeutic index [4,7,8,9,10]. Pharmacokinetic/pharmacodynamic (PK/PD) studies have reported relationships between exposure and treatment outcomes (efficacy and toxicity) for several tyrosine kinase inhibitors (TKI), suggesting a potential interest for drug monitoring [7,11,12,13,14]. Regarding PAZ, a trough plasma concentration (C_min_) ≥ 20.5 mg/L was associated with both improved PFS (19.6 vs. 52.0 weeks, *p =* 0.004) and tumor shrinkage in RCC patients [7]. This efficacy threshold was later confirmed in a real-life RCC cohort [8]. No clear threshold value for PAZ C_min_ was identified for the occurrence of severe toxicity, regardless of the tumor type.

From an exploratory study including 34 STS patients, we previously proposed a PAZ C_min_ threshold of 27 mg/L for efficacy [15]. In the present study, we aimed to confirm this threshold in a larger cohort of unselected STS patients and explore the exposure–response relationship for toxicity.

## 2. Materials and Methods

### 2.1. Study Design and Patients

Between December 2013 and October 2020, all patients with metastatic or unresectable STS or bone sarcoma treated with PAZ in Cochin-Port Royal hospital (Paris, France) were included in this observational study. Patients were considered if at least one plasma concentration of PAZ was available at steady state (after at least 15 days of treatment) (Figure 1). Only STS patients were eligible for the main statistical analysis concerning the exposure–PFS relationship, and for secondary analysis concerning the exposure–OS and exposure–toxicity relationships. Informed consent was obtained from all patients prior to inclusion. The study was approved by the institutional review board for non-interventional research (Approval ID 20210429175029).

### 2.2. Procedures

A comprehensive clinical assessment was systematically performed before treatment initiation. During the treatment period, clinical assessment (e.g., toxicities), as well as biological assessment (blood count and evaluation of liver, renal, and thyroid functions), were performed every 2 weeks for the first 3 months, and then monthly. Patients were instructed to monitor blood pressure at home [16,17]. All adverse events were prospectively graded according to the National Cancer Institute–Common Toxicity Criteria for Adverse Events (NCI-CTCAE) version 5.0 [18]. In case of grade 3–4 toxicity, PAZ was suspended until improvement to grade 1–2. CT scans and/or MRI tumor evaluations were recommended every three months until progression.

The recommended starting dose of PAZ was 800 mg/day. However, for patients with a higher risk of toxicities (Eastern Cooperative Oncology Group-Performance Status (ECOG PS) ≥ 2, age ≥ 75 years old, albumin < 30 g/L, cardiovascular background), a lower starting dose (200 to 600 mg/day) could be prescribed at the discretion of the physician. Subsequently, doses could be adjusted in 200 mg increments or decrements based on tolerance [19].

### 2.3. Pharmacokinetic Assessments

Blood samples were drawn at steady state every two weeks for the first three months and then monthly. Given that the half-life of pazopanib is approximatively 31 h, the steady state was considered to be reached after at least 15 days following either treatment initiation or dosage adjustment [10,20]. Blood was collected in 5-mL lithium heparinized Vacutainer tubes at any time over the administration interval. Samples were centrifuged at 3000 rpm for five minutes at 4 °C, transferred to polypropylene tubes, and kept at −20 °C until assay. Plasma PAZ concentration was measured using high-performance liquid chromatography coupled with UV detection. The calibration was linear in the range 1.2–75 mg/L. The intra- and inter-precision for three internal quality controls (2.5, 14, and 50 mg/L) were below 8 and 10%, respectively. The intra- and inter-accuracy ranged from 92.8 to 109.9%. Finally, the accuracy of the method was ensured by participation in the TKI Proficiency Testing Scheme provided by the Group of Clinical Pharmacology in Oncology (Villejuif, France).

The PAZ C_min_ was estimated using a Bayesian method and the population PK model of Yu et al. [9]. This PK model was developed based on data from 96 patients (31 treated for sarcoma) included in three clinical trials [21,22,23], with doses ranging from 400 to 1200 mg/day. The demographic and biological characteristics of that population were similar to our cohort (median age 53 years old, normal liver, and renal function). Although the population from Yu et al. included more male patients (78%) than our cohort (52%), sex is not a significant covariate on pazopanib PK, as previously reported by Ozbey et al. [10]., which confirmed that the model by Yu et al. could be used to predict PK parameters in our STS patient cohort.

### 2.4. Study Endpoints

Regarding efficacy, the primary endpoint was PFS at 3 months in STS patients. The main objective was to determine whether PAZ C_min_ ≥ 27 mg/L at day 15 (D15) was associated with a longer PFS at 3 months. PFS was calculated as the time between the first day of PAZ to tumor progression or death. Tumor progression was assessed using RECIST 1.1 criteria when measurable lesions were present or was established by the referent oncologist based on clinical findings, with retrospective confirmation by two oncologists (PBR and JA). The cut-off of 3 months for the PFS was chosen for two reasons. First, it corresponded to the first radiologic assessment; second, it was in accordance with the primary endpoint of the PALETTE trial [6].

The secondary endpoints were overall survival (OS) and incidence of dose-limiting toxicity (DLT) during the first three months in STS patients. OS was calculated as the time between the first day of PAZ to death (all causes included). DLT was defined as any clinical or biological grade 3 or 4 toxicity leading to treatment dose reduction, interruption (temporary stop), or permanent discontinuation.

### 2.5. Statistical Analysis

Statistical analyses were performed with the software R (version 4.0.3). Groups were compared with a Student’s t-test for quantitative variables, and Chi2-test for qualitative variables. Based on the results of our exploratory study and the PALETTE trial [6], we calculated that we would need to enroll 82 patients to show a 35% difference of 3-month PFS (70% vs. 35% in patients with a C_min_ ≥ 27 and <27 mg/L, respectively), with a two-sided 5% significance level and an 80% statistical power. Univariate and multivariable logistic regression models were used to test the association of bio-clinical variables with 3-month PFS. Variables associated with significant *p*-value in univariate analyses and potential confounders (initial daily dose, histological subtype) were included in multivariable models, except for tumor grade, as its evaluation is not possible for some histological subtypes. Interaction tests revealed no significant subgroup differences. Survival curves were obtained with Kaplan–Meier estimates and compared with a log-rank test. Univariate and multivariable Cox proportional hazards regression were used to identify variables associated with OS. The proportional hazards assumption was checked for each model using graphical methods based on Kaplan–Meier curves and the scaled Schoenfeld residuals. Univariate and multivariable logistic regression models were used to test the association of bio-clinical variables with DLT. As we focused on PAZ concentration, we also included data on the dosage per kilogram of body weight in each analysis.

All *p*-values were two-sided, and the level of significance was set at *p* < 0.05.

Our cohort study fulfils the STROBE criteria for the reporting of observational studies in epidemiology [24].

## 3. Results

### 3.1. Patients and Treatment

Ninety-five patients with STS were included for statistical analysis. Their baseline characteristics are outlined in Table 1. The median treatment duration was 3.8 months (range 0.4–51.0). At least one dose change was performed in 39 patients (41%), mainly within the first 3 months (38%).

### 3.2. Plasma Concentration of Pazopanib

Four hundred and twenty-six samples were assayed, with a median of three (range 1–20) samples per patient. The target C_min_ ≥ 27 mg/L was not reached in 42% of the samples at the first sampling, and in 38% in the whole cohort. The initial daily dose of PAZ was 800 mg/day in 54 patients (57%). In this subgroup, the inter-individual variability in PAZ C_min_ was 45% at D15. No baseline characteristic was associated with PAZ C_min_ (data not shown). PAZ C_min_ at D15 was significantly higher for patients treated with 800 mg/day compared with those treated at lower dosage (31.5 ± 15.1 vs. 28.5 ± 12.4 mg/L; *p =* 0.03) (Figure 2). Among patients who received 800 mg/day over the whole sampling period (n *=* 17), no variation of PAZ C_min_ over time was observed.

### 3.3. Efficacy of PAZ

The median follow-up was 11.4 months (95% CI 1.4–41.1). At data cut-off, in November 2020, 13 STS patients (14%) were still being treated with PAZ.

The median PFS was 3.3 months (95% CI 2.6–5.1). In univariate analysis, PAZ C_min_ < 27 mg/L at D15 was associated with a higher risk of progression at 3 months (OR 3.09, 95% CI (1.31–7.28), *p =* 0.01) unlike the initial daily dose (Table 2). The multivariate analysis identified PAZ C_min_ < 27 mg/L as an independent risk factor of progression (OR 4.21, 95% CI (1.47–12.12), *p =* 0.008), with other bio-clinical factors (Table 2).

### 3.4. Overall Survival

The median OS was 13.9 months (95% CI 11.4–20.1).

Over the whole follow-up duration, patients with PAZ C_min_ > 27 mg/L at D15 tended to have a longer OS compared to the other patients: 17.7 months (12.0–27.6) vs. 11.4 months (7.1–18.8) (log-rank *p =* 0.07) (Figure 3). In the multivariate analysis, only ECOG PS ≥ 2 was identified as an independent risk factor for OS (HR 2.31 (1.26–4.23), *p =* 0.007) (Table 3).

### 3.5. Relationship between Exposure and Dose-Limiting Toxicity

Overall, 54 DLT events were observed in 39 (41%) patients during the first 3 months of treatment, including hypertension (n *=* 16, 17%), asthenia (n *=* 13, 14%), anorexia (n *=* 11, 12%), and hepatic cytolysis (n *=* 6, 6%). Grade 3–4 toxicities led to dose decrease or treatment interruption for 19 patients (20%) (including 15 patients with an 800-mg initial dose) and 35 patients (37%), respectively. Fourteen patients (15%) were able to have a dose increase within the first three months and a further nine patients (9%) after three months. Baseline characteristics associated with occurrence of DLT are shown in Table 4.

We tested the threshold value of 50 mg/L proposed for DLT onset in RCC patients, but it was not statistically significant (data not shown).

However, a higher average of the three first PAZ C_min_ was associated with a higher risk of grades 3–4 toxicities in the univariate analysis (OR 1.07 per 1 mg/L increase, CI95 (1.02–1.13), *p =* 0.007), with a mean PAZ C_min_ of 39.3 mg/L in patients with DLT vs. 29.6 mg/L in patients with no DLT (*p =* 0.005, Figure 4). Moreover, PAZ C_min_ was an independent predictor of DLT in the multivariable analysis (OR 1.07, CI95 (1.01–1.13), *p =* 0.01) (Table 4).

## 4. Discussion

The pivotal PALETTE trial [6] demonstrated a clinical benefit of PAZ compared to placebo in STS patients treated with 800 mg/day. However, a large interindividual variability in response to PAZ is observed in STS patients in daily clinical practice, both in terms of efficacy and toxicity. Different clinical and biological parameters can contribute to this variability, especially in unselected STS patients treated outside clinical trials. In the present study, different parameters such as PS, tumor grade, bone or node metastasis, BMI, and NLR were identified as risk factors for 3-month progression. These results are in accordance with those of the PALETTE trial.

The main finding of this observational study is that a PAZ C_min_ < 27 mg/L at D15 was an independent risk factor of a lower 3-month PFS, while a starting dose lower than 800 mg/day was not. As in most PK/PD studies [7,25,26,27] for PAZ, we decided to use C_min_ as an exposure parameter because of its significant correlation with AUC_0–24_ [8]. Furthermore, using AUC_0–24_ as a PK parameter in daily clinical practice implies the use of a Bayesian estimator that is not available in most laboratories. A threshold C_min_ of 20.5 mg/L was previously associated with PFS in RCC patients but not in STS patients [7,14]. This discrepancy could be related to the lower efficacy of PAZ in STS patients compared to RCC. We also observed that OS tended to be shorter in patients with PAZ C_min_ < 27 mg/L, but the difference was not statistically significant. In the PALETTE trial, none of the factors explored were found to be significantly associated with OS [28].

The safety profile of PAZ in our real-life cohort is generally consistent with previous studies [5,6]. In the present study, the univariate analysis identified PS and BMI as risk factors of DLT onset. Regarding PK/PD analysis, a higher risk was observed in patients with increased plasma C_min_ over the first 3 months of treatment (OR 1.07 per 1 mg/L increase, *p =* 0.01). In RCC patients (n *=* 205), Suttle et al. showed an increased frequency of hypertension, diarrhea, hepatic cytolysis, and stomatitis in the fourth quartile PAZ C_min_ (36–85 mg/L) [7]. In the present study, we did not investigate any PK/PD relationship for these specific adverse events, because our study was not statistically designed to address this issue. However, we tested the threshold value of 50 mg/L proposed for DLT onset in RCC patients [14,29,30], but it was not statistically significant (data not shown). This threshold is probably less than 50 mg/L in sarcoma patients, owing to their higher fragility compared to RCC patients. Further studies are warranted to clearly identify a threshold value of PAZ C_min_ able to predict DLT onset in sarcoma patients.

The poor tolerance profile of several TKI, especially in patients with poor PS, has led to the evaluation of the use of a lower daily dose at the initiation, with a secondary increase according to tolerance. Such strategies have been validated with regorafenib and afatinib [31,32], but need close clinical monitoring, which is not always feasible in daily practice. The use of therapeutic drug monitoring (TDM) could be an alternative approach to ensure therapeutic plasma exposure over the whole treatment course. In case of suboptimal exposure, a dose escalation strategy should be conducted whenever the safety profile is favorable to it. However, it is noteworthy to underline that a daily dose of PAZ above 800 mg induces saturation of its intestinal absorption [9]. Therefore, Groenland and al. have shown that dividing the dose into two 400-mg intakes is strongly recommended, to enhance the bioavailability in underexposed patients treated with 800 mg/day [19,33].

The major strength of this study is its sample size, which is the largest about PAZ pharmacokinetics in STS. Moreover, both survival and tolerance data from this “real-life” study are consistent with the literature. However, the present study also presents several drawbacks, including the monocentric design and the numerous treatment interruptions within the first three months that could have interfered with PK/PD analysis.

In patients treated with an oral targeted anticancer drug, TDM has been recognized as a powerful tool to individualize drug dosing, ensure drug concentrations within the therapeutic window, and increase treatment success rates [34]. Several PK/PD studies showed the relevance and feasibility of TDM in patients treated with angiogenesis inhibitors, such as sorafenib or sunitinib [12,13,35,36,37]. The present study suggests the clinical benefit of early TDM in STS patients under PAZ, as previously proposed in RCC patients [8]. Some drug–drug interactions are also relevant indications for TDM in STS patients treated with PAZ. For example, coadministration of proton pump inhibitors is known to decrease plasma exposure by 40%, which results in significantly shortened PFS and OS [38]. Thereby, TDM may be helpful for the clinical management of most patients. Overall, early TDM strategy could be helpful to both prevent early treatment failure and DLT onset. According to our results, the targeted range of PAZ concentrations should be 27–40 mg/L in STS patients. This range is narrow; therefore, achieving these therapeutic concentrations may be very difficult in practice, considering the complexity of PAZ PK and its variability. Therefore, regular TDM is recommended, to ensure therapeutic exposure over the entire treatment course.

In conclusion, the present study confirms PAZ C_min_ target > 27 mg/L in a large cohort of STS patients to optimize efficacy. In today’s era of personalized medicine, early TDM could be helpful to optimize the response to PAZ in these patients, as previously reported in RCC patients. In this context, any STS or RCC patient treated with PAZ should have access to TDM.

## Figures and Tables

**Figure 1 pharmaceutics-14-01224-f001:**
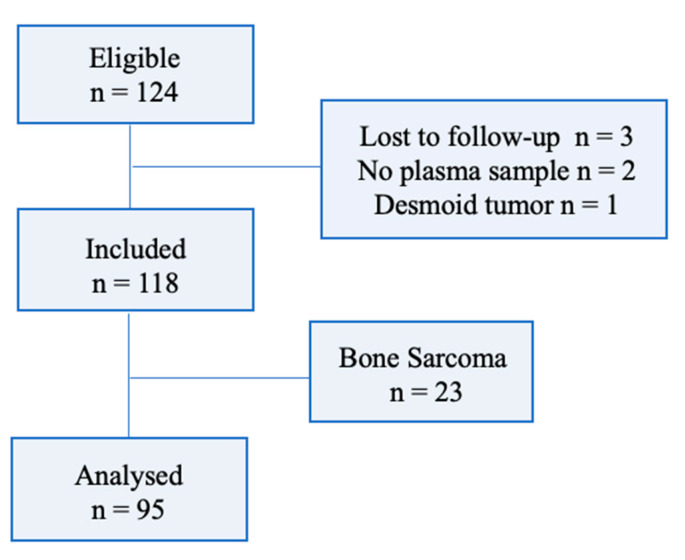
Flowchart.

**Figure 2 pharmaceutics-14-01224-f002:**
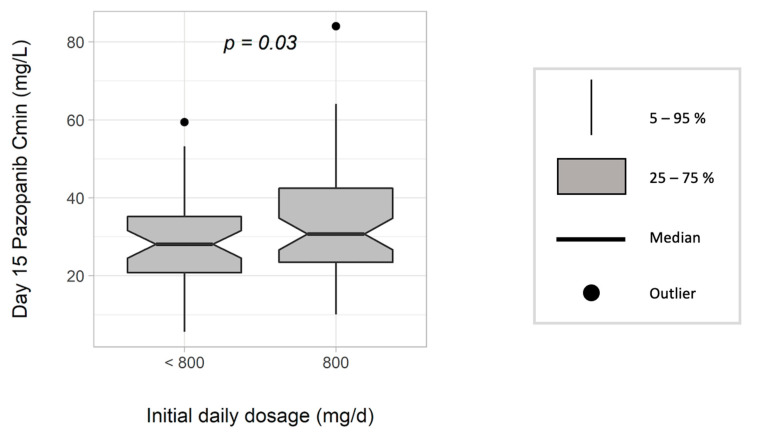
Trough (C_min_) concentration of pazopanib at D15 according to the starting dose (mg/day) (n = 95).

**Figure 3 pharmaceutics-14-01224-f003:**
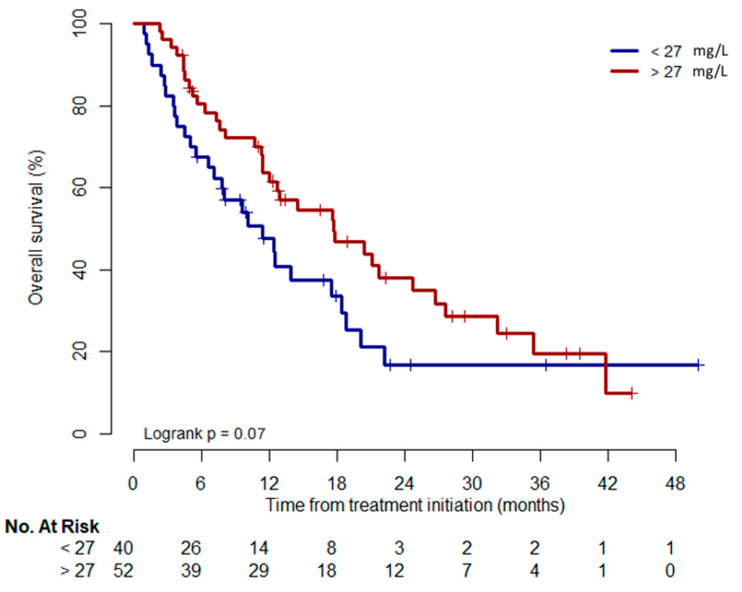
Kaplan–Meier analysis for OS according to pazopanib concentration at D15 (n = 95).

**Figure 4 pharmaceutics-14-01224-f004:**
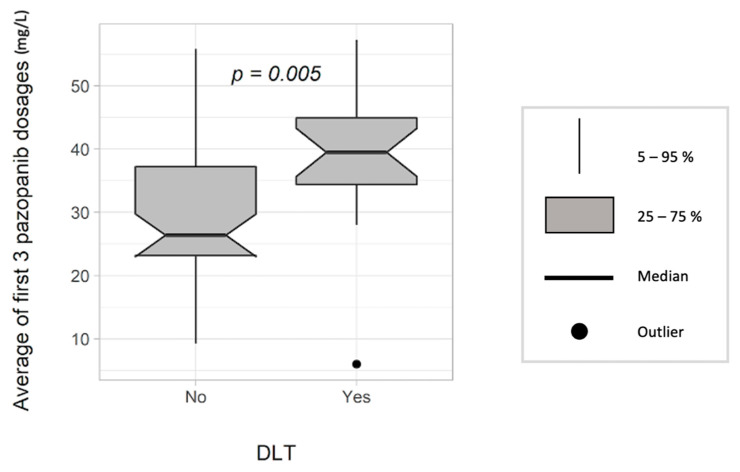
Dose-limiting toxicities (DLT) according to the average of the first three trough (C_min_) concentrations of pazopanib (n = 95).

**Table 1 pharmaceutics-14-01224-t001:** Patient characteristics (n = 95).

Variable	
Sex, n (%) Male	49 (52%)
Median age, years (range)	53.5 (19–83)
Age, n (%) ≥70 years	12 (13%)
Performance status, n (%) 0–1 ≥2	71 (75%) 24 (25%)
Body Mass Index, n (%) <25 ≥25	50 (53%) 45 (47%)
Subtypes, n (%) Leiomyosarcoma Synovial sarcoma Malignant solitary fibrous tumor Epithelioid and clear cell sarcoma Myxofibrosarcoma Undifferentiated pleomorphic and fusiform cell sarcoma Other *	28 (30%) 14 (15%) 12 (13%) 8 (9%) 7 (8%) 6 (6%) 13 (13%)
Site of primitive tumor, n (%) Lower member Upper member Trunk (thorax, abdomen, head and neck) Uterus	43 (45%) 11 (11%) 30 (32%) 11 (11%)
Histological grade, n (%) 1–2 3 Not concerned **	30 (31.5%) 35 (37%) 30 (31.5%)
Pazopanib metastatic line, n (%) First Second ≥Third	20 (21%) 39 (41%) 36 (38%)
Initial daily dose of pazopanib, n (%) 200 mg 400 mg 600 mg 800 mg	2 (2%) 15 (16%) 24 (25%) 54 (57%)
Initial dose-weight, mg/kg (standard deviation)	9.68 (3.1)
Albumin, n (%) <35 ≥35	14 (15%) 81 (85%)
SGOT-SGPT, n (%) ≤ULN (≤40) >ULN (>40)	78 (82%) 17 (18%)
C Reactive Protein, n (%) <10 ≥10	56 (61%) 36 (39%)
Neutrophils/Lymphocytes Ratio (NLR) <3.5 ≥3.5	55 (58%) 40 (42%)

* Including: Malignant solitary fibrous tumor, Epithelioid and clear cell sarcoma, Myxofibrosarcoma, Undifferentiated pleomorphic and fusiform cell sarcoma; Rhabdomyosarcoma, Angiosarcoma, Extraskeletal myxoid chondrosarcoma, Malignant peripheral nerve sheath tumor, Extraskeletal osteosarcoma, Desmoplastic small round cell tumor, dermatofibrosarcoma. ** The grading system does not apply to clear cell sarcoma, alveolar sarcoma, epithelioid sarcoma, desmoplastic small round cell tumor; n: number. mg/kg: milligrams per kilogram. ULN: upper limit of normal.

**Table 2 pharmaceutics-14-01224-t002:** Univariate and multivariable logistic regression models of risk factors for 3-month progression (n = 95).

Variable	Univariate Analysis		Multivariable Analysis	
	**OR [95 CI]**	***p*-Value**	**OR [95 CI]**	***p*-Value**
**Initial daily dose**				
<800 vs. 800 mg	1.95 [0.85–4.5]	0.12	2.15 [0.71–6.57]	0.18
**PAZ C_min_ at D15**				
<27 vs. ≥27 mg/L	3.09 [1.31–7.28]	0.01	4.21 [1.47–12.12]	0.008
**Histological subtype**				
Leiomyosarcoma vs. other	1.22 (0.48–3.07)	0.68	1.99 (0.62–6.40)	0.25
Synovial sarcoma vs. other	0.54 (0.15–1.99)	0.35	0.67 (0.14–3.26)	0.62
**Tumor grade ***				
Grade 3 vs. 1–2	2.91 (1.03–8.20)	0.04		
**Metastatic sites**				
Bone metastasis	2.85 (1.15–7.06)	0.02	2.63 (0.87–7.95)	0.09
Lymph node metastasis	2.78 (1.16–6.70)	0.02	4.55 (1.43–14.46)	0.01
**ECOG PS**				
≥2 vs. 0–1	3.10 (1.16–8.32)	0.02	1.89 (0.55–6.55)	0.31
**Previous treatments**				
≥2 vs. 0–1 previous lines	0.93 (0.40–2.18)	0.87		
**Dosage per kilogram of body weight**				
/1 mg/kg increase	0.88 (0.77–1.02)	0.09		
**BMI**				
>25 vs. ≤25 kg/m^2^	2.63 (1.13–6.13)	0.03	3.05 (1.02–9.15)	0.046
**Albumin**				
≥ 5 vs. <5 g/L	0.77 (0.25–2.41)	0.66		
**NLR**				
≥3.5 vs. <3.5	0.40 (0.17–0.95)	0.04	0.31 (0.10–0.93)	0.04

* Not included in multivariable analysis (tumor grade evaluation is not possible for some histological subtypes) Abbreviations: PAZ C_min_ at D15: pazopanib C_min_ at day 15; PS: Performance Status; BMI: Body Mass Index; NLR: Neutrophil-Lymphocytes Ratio.

**Table 3 pharmaceutics-14-01224-t003:** Univariate and multivariable Cox proportional hazard analysis for OS (n = 95).

Variable	Univariate Analysis		Multivariable Analysis	
	HR (95 CI)	*p*-Value	HR (95 CI)	*p*-Value
**Age**				
≥70 vs. < 70 yrs	1.01 (0.99–1.02)	0.58		
**Sex**				
Female vs. Male	1.02 (0.62–1.68)	0.94		
**Initial daily dose**				
<800 vs. 800 mg	1.38 (0.83–2.30)	0.22	1.01 (0.58–1.77)	0.97
**PAZ C_min_ at D15**				
<27 vs. ≥27 mg/L	1.57 (0.95–2.61)	0.08	1.62 (0.97–2.72)	0.07
**Histological subtype**				
Leiomyosarcoma vs. other	1.08 (0.62–1.87)	0.8	1.19 (0.67–2.13)	0.55
Synovial sarcoma vs. other	0.71 (0.30–1.70)	0.45	0.86 (0.35–2.09)	0.74
**Tumor grade ***				
Grade 3 vs. 1–2	1.72 (0.94–3.18)	0.08		
**Metastatic sites**				
Bone metastasis	1.78 (1.07–2.97)	0.03	1.42 (0.82–2.47)	0.22
Lymph node metastasis	0.98 (0.57–1.67)	0.93		
**ECOG PS**				
≥2 vs. 0–1	2.54 (1.49–4.33)	0.0006	2.31 (1.26–4.23)	0.007
**Previous treatments**				
≥2 vs. 0–1 previous lines	1.44 (0.86–2.41)	0.17		
**BMI**				
>25 vs. ≤25 kg/m^2^	1.41 (0.85–2.33)	0.19		
**Dosage per kilogram of body weight**				
/1 mg/kg increase	0.96 (0.88–1.04)	0.33		
**Albumin**				
≥35 vs. <35 g/L	1.07 (0.53–2.18)	0.85		
**NLR**				
≥3.5 vs. <3.5	1.00 (0.60–1.67)	0.99		

* Not included in multivariable analysis (tumor grade evaluation is not possible for some histological subtypes).

**Table 4 pharmaceutics-14-01224-t004:** Univariate and multivariable logistic regression models of risk factors for dose-limiting toxicities (n = 95).

Variable	Univariate Analysis		Multivariable Analysis	
	OR (95 CI)	*p*-Value	OR (95 CI)	*p*-Value
**Sex**				
Female vs. Male	1.22 (0.54–2.75)	0.64		
**Age**				
≥70 vs. <70 years	1.65 (0.51–5.34)	0.41		
**Initial daily dose**				
<800 vs. 800 mg	0.62 (0.27–1.42)	0.26		
**First 3 PAZ C_min_**				
/1 mg/L increase	1.07 (1.02–1.13)	0.007	1.07 (1.01–1.13)	0.01
**ECOG PS**				
≥2 vs. 0–1	2.88 (1.11–7.51)	0.03	3.53 (0.62–20.25)	0.16
**BMI**				
>25 vs. ≤25 kg/m^2^	2.66 (1.5–6.13)	0.02	1.63 (0.45–5.98)	0.46
**Albumin**				
≥35 vs. <35 g/L	1.44 (0.444.68)	0.54		
**NLR**				
≥3.5 vs. <3.5	1.14 (0.5–2.59)	0.76		
**SGOT or SGPT**				
>40 vs. ≤40	1.39 (0.47–4.09)	0.55		

## Data Availability

Data available within the article.

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
