# Peer review of "Relation between Plasma Trough Concentration of Pazopanib and Progression-Free Survival in Metastatic Soft Tissue Sarcoma Patients"

_pharmaceutics, 2022, doi:10.3390/pharmaceutics14061224_

Round 1

Reviewer 1 Report

The authors investigated the association between trough concentrations and the efficacy and safety. It is interesting and benefit for clinical settings. However, the authors should consider major revise or sub-analysis. PFS and OS analysis test was performed in patients with STS, while toxicity analysis test was performed in patients including bone sarcoma. The authors should examine the efficacy and safety in only patients with STS or both populations.

  1. Introduction: What is a word of ‘TKI’?
  2. Method, Study design and patients: What is the main analysis? Please describe the main and other analysis.
  3. Method, PK assessments: Blood samples were drawn at steady-state. Please describe the timing. Consider adding a figure regarding time after administration and concentration in the section of Result.
  4. Method, Study endpoint: How did you decide the cut-off of 27 mg/L at day 15?
  5. Results: It is difficult to understand the population. So, the authors need to add the number of participants in each figure and table.
  6. Result; BMI was associated with toxicity and 3-month progression, while initial daily dose was not. If the authors focused on the concentrations, dosage per kilogram of body weight need to be examined as well.
  7. Discussion; The authors recommended that 800mg/day divided into twice to enhance the bioavailability. However, the recommendation is impossibly concluded based on the present study.
  8. Discussion: Cmin is associated with the efficacy and toxicity in Pazopanib. And TDM is benefit strategy to optimize the treatment. Then, do you think that how range of concentrations is better?

Author Response

Reviewers' Comments to Author:

The reviewers believe that our manuscript is of potential interest, but they consider that the manuscript requires revision before publication.

We would like to thank the reviewers for their comments that will contribute to a better understanding of the present original research article. Here we have answered and developed each of them.

Changes have been highlighted along the text with track changes, and some extra sentences in order to clarify different aspects.

Reviewer #1 comments:

  1. The authors investigated the association between trough concentrations and the efficacy and safety. It is interesting and benefit for clinical settings. However, the authors should consider major revise or sub-analysis. PFS and OS analysis test was performed in patients with STS, while toxicity analysis test was performed in patients including bone sarcoma. The authors should examine the efficacy and safety in only patients with STS or both populations.

Answer:

We thank the reviewer for this advice to improve our article. We implement this recommendation by removing bone sarcomas from secondary analyses for toxicities. Thus, all the analyses of our article will focus on soft tissue sarcomas only.

Changes:

Page 2, 2.1.Study design and patients, lines 73-74 : Only STS patients were analysed for the main analysis concerning the exposure-PFS relationship and for secondary analysis concerning the exposure-OS and exposure-toxicity relationships.

The flowchart has been updated.

Page 5, 3.Results, figure 1, table 1, figure 2, table 4, and figure 4 have been updated with the new data resulting from the new statistics for soft-tissue sarcoma only.

  1. Introduction: What is a word of ‘TKI’?

Answer:

This point has been modified accordingly.

Changes:

Page 2, line 55 : for several tyrosine kinase inhibitors (TKI)

  1. Method, Study design and patients: What is the main analysis? Please describe the main and other analysis.

Answer:

We thank the reviewer for this remark. We added this information in the part “Materials and Methods”. The main analysis concerns the exposure-PFS relationship and secondary analyses concern exposure-OS and exposure-toxicity.

Changes:

Page 2, 2.1, lines 73-75: Only STS patients were eligible for the main statistical analysis concerning the exposure-PFS relationship and for secondary analysis concerning the exposure-OS and exposure-toxicity relationships.

  1. Method, PK assessments: Blood samples were drawn at steady-state. Please describe the timing. Consider adding a figure regarding time after administration and concentration in the section of Result.

Answer:

We thank the reviewer for this comment.

Given the half-life of pazopanib is about 31 hours, the steady state was considered to be reached after at least 15 days of treatment start or dosing adjustment. We added this sentence in the materials and methods section (lines 101-102). Regarding the supplemental figure, we do not understand whether you want the distribution of all analyzed samples (n=426) over the treatment course (from start to discontinuation treatment) or the 24-hour dosing interval. Therefore, we did not add this figure in the revised manuscript.  

  1. Method, Study endpoint: How did you decide the cut-off of 27 mg/L at day 15?

Answer:

We thank the reviewer for this question. The cut-off of 27 mg/L was obtained from our previous exploratory study (Bellesoeur and al. 2017, doi: 10.1158/1078-0432.CCR-17-1873): it corresponded to the 25th percentile of Cmin,ss in the studied cohort.

  1. Results: It is difficult to understand the population. So, the authors need to add the number of participants in each figure and table.

Answer:

We thank the reviewer for this comment.

This problem has been solved by harmonising the results by keeping only the soft-tissue sarcoma cohort. Thus, each figure and table now present the same number of participants (n=95). Moreover, we systematically added the number of patients in each legend.

  1. Result; BMI was associated with toxicity and 3-month progression, while initial daily dose was not. If the authors focused on the concentrations, dosage per kilogram of body weight need to be examined as well.

Answer:

We thank the reviewer for this comment. This is indeed an interesting complementary information that we added to our article.

Changes:

Page 4 lines 155-156  As we focused on PAZ concentration, we also included data on the dosage per kilogram of body weight in each analysis.

Page 5 table 1 : Initial dose-weight, mg/kg (standard deviation) 9.68 (3.1)

Page 7 table 2: Dosage per kilogram of body weight / 1 mg/kg increase OR 0.88 [0.77-1.02] p=0.09

page 9 table3: Dosage per kilogram of body weight / 1 mg/kg increase HR 0.96 [0.88-1.04] p=0.33

  1. Discussion; The authors recommended that 800mg/day divided into twice to enhance the bioavailability. However, the recommendation is impossibly concluded based on the present study.

Answer:

We thank the reviewer for this remark.

Actually, this recommendation is not based on our study but on the study of Groenland and al. “Boosting pazopanib exposure by splitting intake moments: A prospective pharmacokinetic study in cancer patients.” (doi : 10.1200/JCO.2019.37.15_suppl.3119). We modified the text to make it clearer.

Changes:

Page 12, Discussion, lines 309-310:  Therefore, Groenland and al. have shown that dividing the dose into 400 mg twice is strongly recommended to enhance the bioavailability in underexposed patients treated with 800 mg/day (21,33).

  1. Discussion: Cmin is associated with the efficacy and toxicity in Pazopanib. And TDM is benefit strategy to optimize the treatment. Then, do you think that how range of concentrations is better?

Answer:

We thank the reviewer for this question.

According to our results, the targeted range of PAZ concentrations should be 27-40 mg/L in STS patients. This range is narrow; therefore achieving these therapeutic concentrations could be very difficult in practice regarding to the complexity of PAZ PK and its variability. Therefore, regular TDM is recommended to ensure therapeutic exposure over the entire treatment course. Accordingly, we modified the discussion (lines 330-334).

Reviewer 2 Report

Abstract

Purposes are stated clearly.

Please spell out “PFS” in the methods paragraph.

Please state the grading methodology for adverse effects (CTCAE?).

Please spell out “PK/PD”, “OR”, and “CI” in the results paragraph. PAZ should be abbreviated in the first appearance.

Are 118 patients a large cohort? Or does this fact derive from the previous study? (if yes, it would be better not to mention it in the abstract)

Background (introduction)

This section is stated clearly. No further correction is needed.

Materials and Methods

The subtitle should be added the number (i. e., 2.1)

(Line 92) Please add the explanation for “ECOG PS”

(Lines 99-) For detection methods of PAZ, does this method refer to the previous study? If yes, please add the reference. If not, please add some information for the method validation (accuracy, precision, etc.).

(Line 106) Please state the details of the patients' backgrounds for the established (used) PPK model.

The statistical analysis section was indicated appropriately.

Results.

Please do not embed the Table as a picture (image).

“pazopanib” and “PAZ” are mixed.

Figures 2 and 4: please add the legends for the drawn boxes, dots, and lines.

(Lines 185-) please correct the number of significant digits

(Line 218) “xx (%)” is an inappropriate expression.

(Line 225) “table” > “Table”

“D15” and “day 15” are mixed.

(Line 233) What does “IC95” mean?

(Line 234) “figure” > “Figure”

(Line 275) “TKI” should be spelled out or abbreviated in the first appearance.

(Line 283) “dividing” may be an appropriate expression.

Author Response

Reviewers' Comments to Author:

The reviewers believe that our manuscript is of potential interest, but they consider that the manuscript requires revision before publication.

We would like to thank the reviewers for their comments that will contribute to a better understanding of the present original research article. Here we have answered and developed each of them.

Changes have been highlighted along the text with track changes, and some extra sentences in order to clarify different aspects.

Reviewer #2 comments:

  1. Abstract

Please spell out “PFS” in the methods paragraph.

Please state the grading methodology for adverse effects (CTCAE?).

Please spell out “PK/PD”, “OR”, and “CI” in the results paragraph. PAZ should be abbreviated in the first appearance.

Answer:

We thank the reviewer for these remarks. All these points have been modified accordingly.

  1. Are 118 patients a large cohort? Or does this fact derive from the previous study? (if yes, it would be better not to mention it in the abstract).

Answer:

We thank the reviewer for this question.

We believe that a hundred patients is effectively a large cohort as sarcomas are a rare type of cancers. To compare, in the pivotal study of PK/PD for pazopanib in renal cancers (Suttle and al., 2014), 177 patients were analyzed, while this type of cancer is much more frequent (Incidence in 2018: 15 000 new cases of RCC versus 2700 STS). However, we removed this word in the sentence.

  1. Materials and Methods

The subtitle should be added the number (i. e., 2.1).

Answer:

We thank the reviewer for this remark. All the subtitles have been completed accordingly.

  1. Materials and methods

(Line 92) Please add the explanation for “ECOG PS”.

Answer:

This point has been modified accordingly.

Changes:

Page 3, 2.2, lines 93: ECOG PS (Eastern Cooperative Oncology Group - Performance Status)

  1. Materiel and methods

(Lines 99-) For detection methods of PAZ, does this method refer to the previous study? If yes, please add the reference. If not, please add some information for the method validation (accuracy, precision, etc.).

Answer:

We thank the reviewer for this comment. Given the validation results of this method were not published, we added supplemental data according to your recommendation.

Changes:

Page 3, 2.3, lines 107 to 111:

According to your recommendation, we had supplemental data about the analytical and its performance: “The calibration was linear in the range 1.2-75 mg/L. The intra and inter-precision for three internal quality controls (1.2, 14 and 50 mg/L) was below 8 and 10%, respectively, and the intra and inter-accuracy ranged from 92.8 to 109.9%. Finally, the accuracy of the method was ensured by participation in the TKI Proficiency Testing Scheme provided by the Group of Clinical Pharmacology in Oncology (Villejuif, France).”

  1. Materiel and methods

(Line 106) Please state the details of the patients' backgrounds for the established (used) PPK model

Answer:

We understand the remark of reviewer. This PK model was developed based on data from 96 patients (31 treated for sarcoma) included in three clinical trials with dose ranging from 400 to 1200 mg/day. The demographic and biological characteristics of that population were similar to our cohort (median age 53 years old, normal liver and renal function). Although the population from Yu et al. included more male patients (78%) than our cohort (52%), sex is not a significant covariate on pazopanib PK as previously reported by Ozbey et al. (Pharmaceutics 2021, PMID : 34577627), which confirms that the model by Yu et al. can be used to predict PK parameters in our STS patients cohort.

Accordingly, we modified the manuscript (lines 113-120).

  1. Please do not embed the Table as a picture (image). Modification done.

Answer:

This point has been changed accordingly.

  1. “pazopanib” and “PAZ” are mixed.
  2. “D15” and “day 15” are mixed. Modification done.

Answer:

These two similar points have been changed accordingly.

Changes:

We only kept the term “PAZ” and “D15” by harmonizing.

  1. Figures 2 and 4: please add the legends for the drawn boxes, dots, and lines.

Answer:

We thank the reviewer for this comment.

Changes:

We add a legend concerning each component of the boxplot figure.

  1. (Lines 185-) please correct the number of significant digits.

Answer:

This point has been changed accordingly.

Changes:

Page 6, 3.3, line 204 : The median PFS was 3.3 months

  1. (Line 218) “xx (%)” is an inappropriate expression.

Answer:

This point has been changed accordingly.

Changes:

Page 9, 3.5, line 241

  1. (Line 233) What does “IC95” mean?

Answer:

This point has been changed accordingly. We mean “CI95”.

  1. (Line 225) “table” > “Table”

(Line 234) “figure” > “Figure”

Answer:

This point has been changed accordingly.

  1. (Line 275) “TKI” should be spelled out or abbreviated in the first appearance. Modification done.

Answer:

This point has been modified accordingly.

Changes:

Page 2, line 55 : for several tyrosine kinase inhibitors (TKI)

  1. (Line 283) “dividing” may be an appropriate expression. Modification done.

Answer:

This point has been changed accordingly.

Changes:

Page 12, discussion line 309-310 : Groenland and al. have shown that dividing the dose into 400 mg twice is strongly recommend.

Round 2

Reviewer 1 Report

The authors revised appropriately. No further correction is necessary.

Author Response

Comment: The authors revised appropriately. No further correction is necessary.

Response: We would like to thank once again the reviewer for his/her comments which will contribute to a better understanding of this original research article. 

Reviewer 2 Report

We do not believe that a major revision is necessary. However, please check the abbreviations and document style carefully during document proofreading.

Author Response

Comment: We do not believe that a major revision is necessary. However, please check the abbreviations and document style carefully during document proofreading.

Response: We would like to thank once again the reviewer for his/her comments which will contribute to a better understanding of this original research article.
We have made some changes to the sentence structure to improve the style after careful proofreading by a colleague who is very fluent in English.  Moreover, all abbreviations have been checked.